# Integrative Multimodal Deep Learning for Individualized Recurrence Risk Stratification in Stage I–III Colon Cancer

**Melissa de Bruin**[1]                                                    M.D.BRUIN@NKI.NL
**Myriam Chalabi**[1]                                                      M.CHALABI@NKI.NL
**Xinglong Liang**[*1]                                                     X.LIANG@NKI.NL
**Milan Pijl**[4]                                                          MPIJL@RIJNSTATE.NL
**Geerard Beets**[3,5]                                                     G.BEETS@MUMC.NL
**Tianyu Zhang**[†1,2,3]                                                   T.ZHANG@NKI.NL
**Regina Beets-Tan**[2,5]                                                  R.BEETSTAN@MAASTRO.NL

[1] *Netherlands Cancer Institute, Amsterdam, The Netherlands*

[2] *MAASTRO Radiotherapy Clinic, Maastricht, The Netherlands*

[3] *Maastricht University Medical Center+, Maastricht, The Netherlands*

[4] *Rijnstate Hospital, Arnhem, The Netherlands*

[5] *Maastricht University, Maastricht, The Netherlands*

## Abstract

Precise risk stratification in stage I–III colon cancer remains a clinical challenge, as conventional radiological staging often fails to identify high-risk patients. This study developed a multimodal deep learning model integrating preoperative CT imaging with clinical data to predict recurrence. In a retrospective cohort of 713 patients from two centers, the model utilized a convolutional neural network with an attention mechanism for image feature extraction and a multilayer perceptron for clinical variable processing. A late fusion strategy was employed to generate a unified risk score. The model achieved a C-index of 0.665 and a 2-year recurrence AUC of 0.766. Stratification by median risk score yielded a significant hazard ratio of 2.46. These findings suggest that integrating heterogeneous data sources through deep learning provides a robust prognostic tool, potentially facilitating personalized therapeutic interventions.

**Keywords:** Multimodal Deep Learning, Colon Cancer Recurrence, Contrastive Pre-training, Attention Mechanism, Prognostic Stratification.

## 1. Introduction

Accurate risk stratification in stage I–III colon cancer is a significant clinical challenge, as many patients experience recurrence despite curative-intent treatment. While the FOx-TROT trial established the role of neoadjuvant chemotherapy, identification of high-risk candidates currently relies on CT-based staging, which suffers from inherent diagnostic limitations (FOxTROT Collaborative Group, 2012; van den Berg et al., 2020). Meta-analyses indicate suboptimal CT performance in detecting tumor invasion $\geq 5$ mm beyond the bowel wall (77% sensitivity, 70% specificity) and limited accuracy in nodal staging, a key predictor of relapse (Nerad et al., 2016; Dighe et al., 2012).Conventional radiological assessment is further constrained by its dependence on predefined, qualitative semantic features, which fail

---

* Correspondence: X.L

† Correspondence: T.Z.

to capture the high-dimensional morphological and spatial complexity found in CT imaging. This limitation leads to significant prognostic heterogeneity among patients with identical radiological stages (Kang et al., 2022; Dienstmann et al., 2017). To overcome these barriers, multimodal deep learning architectures can integrate volumetric imaging features with structured clinical data. By synthesizing these heterogeneous data sources, such models may better reflect underlying tumor biology and provide a more robust framework for recurrence prediction. This study evaluates a multimodal model designed to refine risk stratification within a time-to-event framework, aiming to improve individualized treatment strategies beyond standard radiological staging.

## 2. Materials and Methods

Study Cohort: A total of 713 patients with histologically confirmed stage I–III colon cancer were retrospectively identified from The Netherlands Cancer Institute and Rijnstate Hospital. Inclusion criteria required a staging CT scan performed within 45 days prior to the primary surgical resection. Clinical data and long-term follow-up records were extracted to determine the time to recurrence.

The proposed framework employs a dual-stream multimodal architecture designed for high-dimensional feature alignment (Figure 1). (1) Imaging Module and Contrastive Pre-training: The imaging stream utilizes a 3D CNN backbone optimized via vision-language contrastive pre-training. By training on paired CT volumes and unstructured radiological reports, the encoder was calibrated to maximize cosine similarity between image embeddings and textual descriptions in a shared latent space. This process ensures the extraction of domain-specific semantic features linked to tumor aggressiveness. (2) Spatial and Channel Attention (SCA): To refine feature localization, the encoder integrates an SCA mechanism. The spatial component generates a probability map to prioritize the primary tumor and peritumoral regions, while the channel component recalibrates feature maps to emphasize discriminative filters associated with factors such as lymphovascular invasion. (3) Multimodal Fusion: Structured clinical variables are one-hot encoded and processed through a Multilayer Perceptron (MLP). Latent representations from both the imaging and clinical streams are integrated using a late fusion strategy. The resulting joint embedding is processed by fully connected layers to output a continuous recurrence risk score, facilitating precise time-to-event risk stratification.

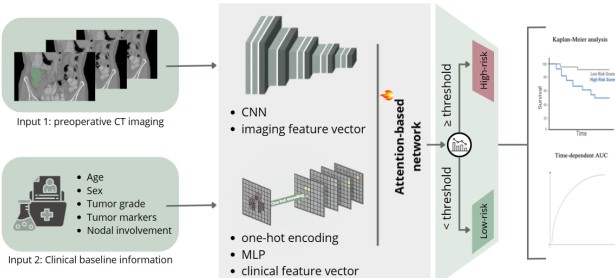

Figure 1: Workflow of this study.

Statistical Analysis: The primary endpoint was recurrence-free survival. The prognostic capability of the risk score was quantified using the concordance index (C-index). To eval-

uate longitudinal performance, the area under the receiver operating characteristic curve (AUC) was calculated specifically for 2-year recurrence. Patients were categorized into low- and high-risk groups based on the median risk score of the cohort. Survival differences between these groups were analyzed using Cox proportional hazards models to derive hazard ratios (HR) with 95% confidence intervals (CI). Significance was defined at $p < 0.05$.

## 3. Results

The integrative multimodal model demonstrated superior prognostic performance compared to single-modality approaches. For the primary endpoint of recurrence-free survival, the multimodal model achieved a C-index of 0.665 (95% CI: 0.576–0.739), significantly outperforming models based solely on imaging (C-index = 0.622) or clinical data (C-index = 0.615). As shown in Figure 2(A), when the cohort was dichotomized by the median risk score, the high-risk group exhibited a significantly higher probability of recurrence with a hazard ratio (HR) of 2.46 (95% CI: 1.31–4.62, $p < 0.01$).

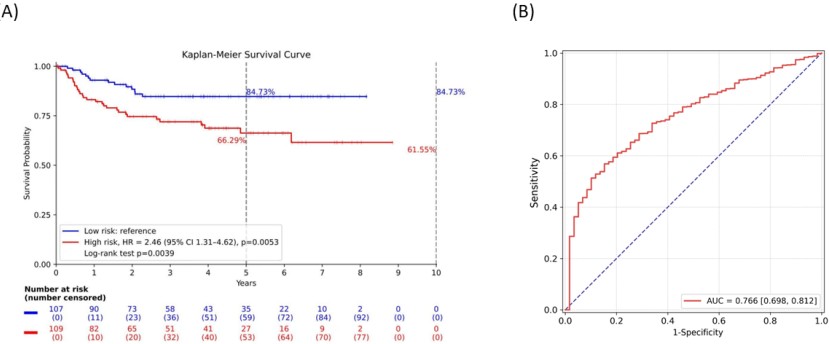

Figure 2: Performance Analysis. (A) Kaplan–Meier analyses. (B) ROC curve

Furthermore, for binary recurrence prediction within a 2-year horizon, the multimodal model yielded an AUC of 0.766 (95% CI: 0.698–0.812) (Figure 2(B)), which was consistently higher than the imaging-only (AUC = 0.732) and clinical-only (AUC = 0.718) models. These results indicate that the fusion of CT-derived spatial features and clinical parameters provides a more sensitive measure of biological aggressiveness than traditional markers. Additionally, the attention mechanism ensured performance stability across disparate CT protocols, demonstrating robustness to multi-center data variability.

## 4. Conclusion

This study demonstrates that a multimodal deep learning approach significantly enhances risk stratification in stage I–III colon cancer compared to conventional staging. By integrating deep imaging features with structured clinical data, the model provides a quantitative risk score that correlates strongly with recurrence outcomes. This methodology effectively captures the interplay between tumor morphology and clinical context, offering a potential tool to refine decision-making for intensive surveillance or adjuvant therapy. Future research will focus on external validation in prospective cohorts and the incorporation of genomic data to further augment its predictive power.

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
