# OpenReview forum: "Integrative Multimodal Deep Learning for Individualized Recurrence Risk Stratification in Stage I–III Colon Cancer"
_MIDL.io/2026/Short_Papers — MIDL 2026 - Short Papers Poster_

### Official Review · Reviewer_t5tq · 2026-04-23
**Sound Method, weak results**

**Rating:** 3
**Confidence:** 4

**Review:**

The authors propose a straightforward architecture for making predictions based on multimodal data. However, certain details are missing, and the claim of superiority compared to prediction over the predictions based on single modality is very weak. I would expect a stronger numerical result or a very good justification to accept the evidence for the claim made.

**Summary:**

The authors present a multimodal model for predicting the recurrence risk of colon cancer. The model consist of a 3D-CNN for embedding images into a latent space, an MLP for the non-imaging data and a network merging these features. The main claim of the paper is that using the two different types of data jointly results in a better performance of the classification.

**Strengths:**

The paper is well written and easy to understand. The method that the authors presented is plausible and clearly illustrated by the corresponding graphics. The results are clearly communicated and illustrated.

**Weaknesses:**

There is some information missing that would be important to interpret the results. On the one hand some data about the training and pretraining, including size and distribution of the test set  as well as distribution of the underlying data would be helpful. The justification for the claimed “superior prognostic performance” compared to the single-modality classifiers is rather weak: The authors report an AU-ROC of 0.766 (with a 95% confidence interval of 0.698-0.812) compared to 0.732 and 0.718 for image-only and clincial-data-only respectively. For interpreting these numbers better, the distribution of the test set would be necessary, and it would be nice to include the ROC-curves for the single-modality classifiers in the Figure 2.B.

**Justification Of Rating:**

The presented method is sound, but the result presented in the evaluation is not convincing.

---

### Decision · Program_Chairs · 2026-05-08

Accept (Poster)